# Relationships of Whole-Heart Myocardial Mechanics and Cardiac Morphometrics by Transthoracic Echocardiography with Main Prognostic Factors of Heart Failure in Non-Ischemic Dilated Cardiomyopathy

**DOI:** 10.3390/jcm12062272

**Published:** 2023-03-15

**Authors:** Karolina Mėlinytė-Ankudavičė, Eglė Ereminienė, Vaida Mizarienė, Gintarė Šakalytė, Jurgita Plisienė, Vytautas Ankudavičius, Rūta Dirsienė, Remigijus Žaliūnas, Renaldas Jurkevičius

**Affiliations:** 1Department of Cardiology, Medical Academy, Lithuanian University of Health Sciences, LT-44307 Kaunas, Lithuania; 2Institute of Cardiology, Lithuanian University of Health Sciences, LT-50162 Kaunas, Lithuania; 3Department of Pulmonology, Medical Academy, Lithuanian University of Health Sciences, LT-44307 Kaunas, Lithuania

**Keywords:** non-ischemic dilated cardiomyopathy, myocardial mechanics, heart failure, 2D echocardiography

## Abstract

Background: there are many prognostic factors of heart failure (HF) based on their evaluation from imaging, to laboratory tests. In clinical practice, it is crucial to use widely available, cheap, and easy-to-use prognostic factors, such as left ventricular ejection fraction (LVEF), New York Heart Association (NYHA) functional class, 6 min walk test (6MWT), B-type natriuretic peptide (BNP), etc. We sought to evaluate the relationships between whole-heart myocardial mechanics and cardiac morphometrics with the main commonly used prognostic factors of HF in patients with non-ischemic dilated cardiomyopathy (NIDCM). Methods and results: two-dimensional (2D) echocardiography for myocardial mechanics (global longitudinal, radial, and circumferential strains of the left ventricle; right ventricular longitudinal strain; strain values of reservoir, conduit, and contraction function of both atria) and cardiac morphometric (diameters and volumes of both atria and ventricles) parameters were performed, and the HF main traditional prognostic factors were identified. We assessed 109 patients (68.8% male; 49.7 ± 10.5 years) with newly diagnosed NIDCM. Myocardial mechanics and morphometrics were weakly correlated with the patient’s age, gender, and smoking (R = 0.2, *p* < 0.05). Stronger relationships were observed with NYHA class, 6MWT, and BNP (the strongest correlations were with LVEF: R = −0.499, R 0.462, R = −0.461, *p* < 0.001, respectively). There were moderately strong correlations with LVEF and other whole-heart myocardial mechanics or morphometrics. Moreover, LVEF with global regurgitation volume (GRV) and right ventricle free wall longitudinal strain (RVFWLS) were the most usually detected parameters in multivariate analysis to be associated with changes in HF prognostic factors. Conclusions: in NIDCM patients, the main prognostic factors of HF are correlated with whole-heart myocardial mechanics and morphometrics. However, LVEF, GRV, and RVFWLS are the most usually found 2D echocardiographic factors associated with changes in HF prognostic factors.

## 1. Introduction

Dilated cardiomyopathy (DCM) is a myocardial disease that causes cardiac dysfunction and HF. Patients with NIDCM have dilatation and systolic dysfunction of left or both ventricles, without coronary artery disease and abnormal loading conditions [1,2,3,4]. Determination of the factors associated with HF severity is important to improve prognostication, and develop more effective therapeutic and diagnostic strategies in HF patients [4]. Poor outcomes in NIDCM are related to a lower LVEF and 6MWT, higher NYHA class or natriuretic peptide concentration, older age, and male gender [4,5,6,7].

Two-dimensional echocardiography is the first-line imaging test in the assessment of patients with DCM. The evaluation of myocardial mechanics by speckle-tracking echocardiography (STE) in NIDCM plays an important role in HF, and may add additional value to improve risk stratification with other clinical markers or biomarkers [8]. Myocardial mechanics are expressed as a strain, which is a parameter that indicates the change in length of a myocardial segment associated with its initial length, and is expressed as a percentage (%) [9]. DCM is related to reduced myocardial mechanics of both atria and ventricles. Significantly reduced strain in all directions is associated with faster progression of HF [9]. However, the progression of the disease is not only associated with changes in myocardial mechanics. DCM is usually related to the enlargement of both ventricles and atria. The geometric changes of the cardiac chambers are associated with disease prognosis, as well as changes in myocardial mechanics, and could be useful in assessing therapeutical response [10].

More data are needed to evaluate which whole-heart myocardial mechanics and cardiac morphometric parameters are associated with the main prognostic factors of HF. In this context, our study objective was to evaluate the potential relationships between myocardial mechanics and morphometric parameters of both ventricles and atria with the main prognostic factors of HF, in a prospective cohort of NIDCM patients.

## 2. Materials and Methods

### 2.1. Study Population

The patients were studied at the Hospital of Lithuanian University of Health Sciences Kaunas Clinics. DCM definition was established according to the current criteria as left ventricle (LV) or both ventricle dilatation and systolic dysfunction (LVEF ≤ 50%), without coronary artery disease (CAD) [2]. CAD was ruled out by coronary angiography (>50% stenosis in one or more coronary arteries). The study included ambulatory and hospitalized patients diagnosed with NIDCM for the first time (patients without chronic or worsening HF). Patients with significant valve disease, inflammatory myocardial disease, or kidney disease (eGFR < 30 mL/min/1.73 m^2^), tachycardia-induced HF (chronic or prolonged unknown duration tachysystolic form of atrial fibrillation or atrial flutter), peripartum cardiomyopathy, with the implantation of an intra-cardiac defibrillator or cardiac resynchronization therapy, in the case of alcohol or drug abuse, and under the age of 18, were excluded from the study.

The initial examination of the patients was based on a detailed medical history (symptoms duration, medications, family history, etc.), physical examination, laboratory tests, 12-lead baseline electrocardiogram, 2D transthoracic echocardiography, and Holter monitoring (detection of rhythm disorders such as ventricular tachycardia, paroxysmal atrial fibrillation, etc.).

The study received institutional ethical approval and all patients provided written informed consent.

### 2.2. 2D Echocardiography Analysis

Two-dimensional echocardiography was performed using Philips “EPIQ 7 ultrasound system by one experienced echocardiographer, and stored images were analyzed offline (TomTec Imaging Systems, Unterschleissheim, Germany). Echocardiography was performed during the first contact with the patient within 24 h of the start of hospitalization, or if it was an outpatient, during the first visit.
Global regurgitation volume

The GRV was expressed as the sum of mitral and tricuspid regurgitant volumes, using the proximal flow convergence method. Three consecutive beats were average in the sinus rhythm and five consecutive beats in atrial fibrillation [11] (Figure 1).
Left ventricle

The LV end-systolic and end-diastolic diameters were obtained from parasternal LV long-axis view, and measured below the level of the mitral valve leaflet tips. LV volumes were obtained from the apical four- and two-chamber views. The volumes were calculated by the biplane method of disk summation. LVEF was calculated by the Simpson’s biplane method [12].

For the evaluation of LV global longitudinal strain (LVGLS), apical four-chamber, two-chamber, and long-axis views were obtained. LV global circumferential strain (LVGCS) and global radial strain (LVGRS) were calculated by endocardial tracing in the basal, middle, and apical levels of LV short-axis views [13].
Right ventricle

The right ventricular (RV) dimensions were estimated from an RV-focused apical four-chamber view [12]. Global RV longitudinal strain (GRVLS) was calculated by averaging peak strain values of six segments (three from the RV-free wall and three from the interventricular septum). RV-free wall longitudinal strain (RVFWLS) was calculated from the RV-free wall segments [14] (Figure 2a).The right and left atria

The left atrial (LA) size was measured at the end of the LV systole (when the LA chamber is at its greatest diameter), and the LA volume was assessed in apical four-and two-chamber views, using the disk summation algorithm. The right atrial (RA) volume was measured using a single-view area-length technique [12]. The single apical four-chamber view was used to automatically evaluate both atria mechanics during the reservoir, conduit, and contraction phases. (Figure 2b,c) [14].

### 2.3. Statistical Analysis

Patients’ characteristics were provided by n (%), or the means± standard deviations (SD). The Kolmogorov-Smirnov test was used to evaluate the normality distribution of the data. Pearson’s correlation coefficient was used to evaluate correlations between prognostic factors of HF with myocardial mechanics and morphometrics. If one variable was categorical and one was continuous, the point-biserial coefficient of correlation R^2^ was calculated. Multivariate regression analysis was used to determine which myocardial strain and morphometric parameter were most associated with prognostic factors of HF. First, for the selection of myocardial deformation and morphometric values, univariate analysis was performed. Univariate analysis was followed by stepwise multivariate linear regression and standardized coefficients (ß), and 95% confidence intervals (CI) were obtained. The data were analyzed using SPSS version 22 (IBM, Chicago, IL, USA). A *p*-value < 0.05 was considered significant.

Twenty-five patients were randomly selected to evaluate the intra-observer variability (Bland–Altman analysis was done), which showed good agreement, with the small bias of 0.7 ± 2.7%.

## 3. Results

Table 1 represents the demographic and clinical characteristics of the study group. The study group consisted of 109 patients with NIDCM. The average age in the NIDCM group was 49.7 ± 10.5, and there were more males (75 (68.8%)). The study group tended to be overweight (body mass index 29.1 ± 5.7). The mean systolic blood pressure and heart rate were within normal limits. AF was present in 39.4%, and ventricular tachycardia was detected in 30.3% of the patients. The patients with NIDCM had a wide QRS duration (approximately 45% of the patients with NIDCM had a left bundle branch block). There were more patients with NYHA functional classes III–IV (56.9% and 13.8%, respectively). Almost half of the patients had a family history of cardiovascular diseases (44.0%). The majority of patients had significantly decreased LVEF (92 (84.4%)). Each patient with NIDCM had more than one risk factor, such as arterial hypertension (AH), dyslipidemia, diabetes, smoking, or obesity (more than half of the patients were obese and had arterial hypertension). During the initial contact, the indicated drugs were usually used for AH alone or in combination therapy. The biomarkers of HF (troponin I, B-type natriuretic peptide, high-sensitivity C-reactive protein) were elevated.

The patients with NIDCM had dilated LV and both atria. It was evaluated by measurement of the diameters and volumes of all chambers. A significant reduction of LVEF was detected (27.7 ± 8.7). Moreover, all myocardial mechanical parameters of both ventricles and atria were reduced. There were no cut-off limits for GRV severity; however, our study revealed that the mean GRV was increased (49.5 ± 32.8) (Table 2).

Table 3 represents whole-heart myocardial mechanics and cardiac morphometric parameter correlations with prognostic factors of HF. There were not any correlations between myocardial mechanics and cardiac morphometric parameters with TnI, diabetes mellitus, heart rate, chronic kidney disease, or hemoglobin. Therefore, these prognostic factors of HF are not presented in the table. Since LVEF is one of the most usual and widely evaluated and studied traditional prognostic factors of HF, it was evaluated both as a parameter of LV mechanics and as a prognostic factor. Myocardial mechanics and morphometrics were weakly correlated with the patient’s age, gender, and smoking (R = 0.2, *p* < 0.05). There were weak–moderate correlations between systolic blood pressure and the volumes of LV and GRV. AF was related to changes in LA mechanics and LA volume index (*p* < 0.001). Stronger relationships were observed between myocardial mechanics and morphometrics with NYHA class, 6MWT, and BNP (the strongest correlations were with LVEF: R = −0.499, R 0.462, R = −0.461, *p* < 0.001, respectively). There were moderate–strong correlations with LVEF and other whole-heart myocardial mechanics or morphometrics.

The systolic blood pressure, NYHA class, 6MWT, BNP, and LVEF were included in multivariate regression analysis (Table 4). The results showed which myocardial mechanical and cardiac morphometric parameters were independently associated with the mentioned prognostic factors of HF. The systolic blood pressure was best correlated with LV end-systolic volume index (*p* = 0.013). NYHA class was associated with changes in GRV (*p* = 0.018), LASr (*p* = 0.006), LVEF (*p* < 0.001), and RVFWLS (*p* = 0.049). A similar tendency was noticed in correlation with 6MWT. BNP concentration was most associated with LVEF. The systolic LV function correlated well with all mechanical and morphometric parameters; however, the strongest correlations were with LVGLS, LVGRS, GRV, and LASr.

## 4. Discussion

Our study encourages simple non-invasive clinical tools to evaluate the relationships between the main prognostic factors of HF and whole-heart myocardial mechanics and morphometrics. The diagnosis of NIDCM includes various diagnostic methods, but echocardiography remains the cornerstone. This study focused on the whole-heart myocardial mechanistic and geometric insights of NIDCM for a better understanding of pathophysiology relationships with prognostic factors of HF, with the purpose of showing some possible new imaging markers through echocardiography.

The findings of this study can be summarized as follows: (1) in patients with NIDCM, the prognostic factors of HF are correlated with the whole-heart myocardial mechanics and cardiac morphometric parameters; (2) LVEF, GRV, and RVFWLS are the most usually found 2D echocardiographic factors associated with changes of HF prognostic factors.

Prognostic factors of HF should be evaluated early [15,16]. It is very important to discuss with the patients their disease prognosis or possible causes of mortality because this discussion is important to help guide treatment and allow for patient-centered advanced care planning [17]. Investigation of whole-heart myocardial mechanics and morphometrics is essential in assessing the function of the entire heart in patients with NIDCM, since this group of patients has a high risk of HF progression due to both ventricles and atria pathological remodeling. To the authors’ knowledge, no study has specifically addressed evaluating the relationships between whole-heart myocardial mechanics and morphometric parameters with prognostic factors of HF in NIDCM.

Many studies analyzed the independent predictors of HF severity and outcomes [18,19]. Prognostic factors of HF include LVEF, NYHA class, older age, male sex, low exercise capacity (6MWT), low systolic blood pressure, low hemoglobin, etc. [19,20,21]. Several clinical prediction models have been created for HF (The Seattle Heart Failure Model, etc.) [18,22]. However, the appearance of these models in DCM is not accurate, because these models include patients with different etiology of DCM. Our study group consisted of selected patients with a non-ischemic origin of DCM. In our study, NYHA class, 6MWT, BNP, and LVEF were the main prognostic factors of HF, most associated with various parameters of myocardial mechanics and cardiac morphometrics. As LVEF is the most studied, the expediency of its monitoring in HF is clear, since reduced LVEF is associated with worse prognosis, increased mortality, and hospitalization rates [23,24,25,26]. Though many studies have previously shown that LVGLS has been proposed as a more sensitive indicator of abnormal systolic function than LVEF [27,28], there were better correlations between myocardial mechanics and morphometrics with LVEF compared to LVGLS. BNP is one of the most common HF indicators [29]. Some studies revealed that biomarkers such as BNP have a good prognostic significance in the diagnosis of HF [30]. Cho et al. identified that NT-proBNP and LV size were independent predictors of LV functional recovery [31]. The knowledge of myocardial mechanical parameters that can be associated with prognostic factors of HF could help to predict the LV functional recovery or outcomes of HF, using only 2D echocardiographic parameters.

Despite age being one of the strongest parameters affecting mortality risk in HF [15,32,33], there was no significant correlation between age and myocardial mechanics or morphometrics in our study. A similar tendency was noticed with gender (only a weak correlation was noticed with LVGRS and RVESVi). AF is another important prognostic factor associated with HF severity in many previous studies [34,35]. AF can be a cause or consequence of HF, however, our group consisted of patients without arrhythmogenic origin of NIDCM. Our results show that AF was related to alterations of LA myocardial mechanics. This confirmed the fact that increased filling pressures and afterload may lead to increased LA stretch and maintenance of AF in HF [34].

Focusing on either valve separately, it does not reflect on the global hemodynamic burden arising from concomitant functional regurgitation of the mitral and tricuspid valves [11]. Bartko et al. presented the first large-scale study about the negative effects of outcomes associated with GRV among patients with reduced LVEF. They declared that the threshold where HF is worsened by the valve lesions is a GRV of 50 mL [11]. There are no studies to describe GRV relationships with prognostic factors of HF. This is a new echocardiographic parameter and we evaluated it in our research to find correlations between GRV with the most important prognostic factors of HF. We have found that GRV was one of the most usually detected 2D echocardiographic factors associated with changes in NYHA class, 6MWT, and LVEF.

RV dysfunction has a known crucial role in functional capacity and prognosis in HF, regardless of the degree of LV dysfunction. The issue of which factors contribute to RV dysfunction in HF is still unclear. In our study, patients with NIDCM had decreased mechanical strain parameters of RV. It was found that RVFWLS is well correlated with NYHA class and 6MWT [36].

The patient’s functional status (NYHA class, 6MWT), instrumental and laboratory tests were important in daily clinical practice to follow the dynamics of the patient’s disease, the effectiveness of the treatment and to determine the indications for the implantable devices. This monitoring is important for a timely referral to specialized HF centers. Echocardiography is the most important non-invasive instrumental tool in the care of HF patients. The results of this study showed that the main prognostic factors of HF correlate with whole-heart myocardial mechanics and morphometric parameters. Regardless of LVEF, GRV and RVFWLS were the other most usually found 2D echocardiographic factors associated with changes in HF prognostic factors. Based on this, this might aid additional values in the clinicians’ decision-making, and the reclassification of patients suitable for interventional treatment or assessing the effectiveness of medical treatment.

## 5. Limitations

There are several limitations to our study. This is a single-center experience and is related to the small sample size. Additionally, our study group may not represent all patients with NIDCM. We evaluated only the main and most common prognostic factors of HF in clinical practice. Our study revealed weak–moderate correlations between prognostic factors of HF and myocardial mechanics or morphometrics of both ventricles and atria. However, these correlations were statistically significant. Further research is needed to confirm our findings. The quantitative assessment of the whole-heart structure and function, especially the assessment of RV and right atrium, is challenging. However, the limited access to the technology in daily practice and common contraindications in the presence of implanted devices make other technologies, especially cardiac magnetic resonance imaging, less suitable than 2D echocardiography for studies involving large cohorts of patients. Based on this, we presented only 2D echocardiography for the assessment of myocardial mechanics and morphometrics.

## 6. Conclusions

In NIDCM patients, the main prognostic factors of HF are related to the whole-heart myocardial mechanics and morphometrics. Combined with LVEF, GRV and RVFWLS may add additional clinical value to patients selected for advanced treatment or assessing the effectiveness of medical treatment.

## Figures and Tables

**Figure 1 jcm-12-02272-f001:**
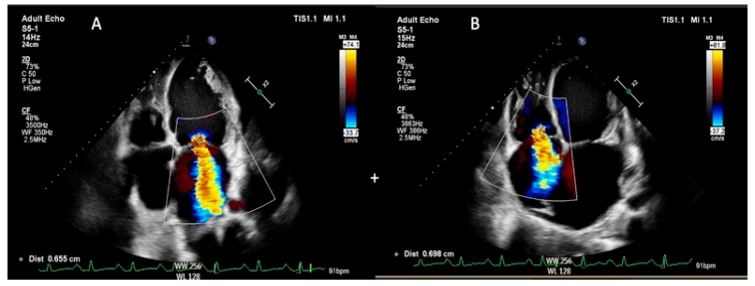
Global regurgitation volume estimation ((**A**)—mitral regurgitation and (**B**)—tricuspid regurgitation; the quantitative assessment was performed by the proximal flow convergence method).

**Figure 2 jcm-12-02272-f002:**
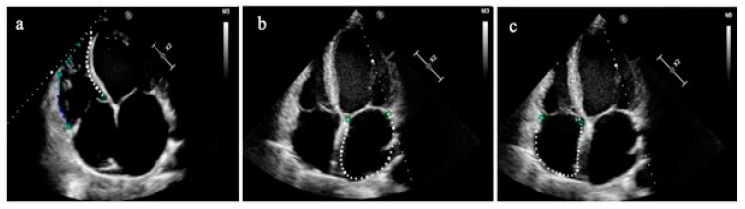
(**a**) Right ventricular speckle-tracking analysis (global RV longitudinal strain is indicated by a white and colored dotted line; RV-free wall longitudinal strain is indicated only by a colored dotted line); (**b**,**c**) Left and right atrial strains are composed of six segments and indicated by a whited dotted line, respectively.

**Table 1 jcm-12-02272-t001:** Patients’ demographic and clinical characteristics.

Demographic and Clinical Variables	NIDCM Group (n = 109)
Age, years	49.7 ± 10.5
Male gender, n (%)	75 (68.8)
BMI, kg/m^2^	29.1 ± 5.7
Systolic blood pressure, mmHg	125.4 ± 13.8
Heart rate, bpm	75.0 ± 8.1
Atrial fibrillation, n (%)	43 (39.4)
QRS duration, ms	120.9 ± 29.6
NYHA class, n (%)	
I	3 (2.8)
II	29 (26.6)
III	62 (56.9)
IV	15 (13.8)
HFrEF (≤40), n (%)	92 (84.4)
HFmrEF (41–49), n (%)	17 (15.6)
Positive family history, n (%)	48 (44.0)
VT, n (%)	33 (30.3)
Cardiovascular risk factors, n (%)	
Arterial hypertension	56 (51.4)
Dyslipidemia	45 (41.3)
Diabetes	8 (7.3)
Smoker	44 (40.4)
Obesity	70 (64.2)
Pharmacotherapy (at baseline), n (%)	
ACE-I/ARB	43 (39.4)
Betablocker	51 (46.7)
CCB	28 (25.6)
Aldosterone antagonist	9 (8.2)
Statins	20 (18.3)
Diuretic	3 (2.7)
Laboratory values	
Hgb, g/L	139.8 ± 13.3
TnI, ng/L	0.3 ± 1.2
hs-CRP, mg/L	3.2 ± 2.8
BNP, ng/L	1256.3 ± 680.3

NIDCM—non-ischemic dilated cardiomyopathy; BMI—body mass index; QRS—QRS complex on the electrocardiogram; NYHA—New York Heart Association; VT—ventricular tachycardia; ACE-I—angiotensin-converting enzyme inhibitor; ARB—angiotensin receptor blocker; CCB—calcium channel blocker; Hgb—hemoglobin; TnI—troponin I; BNP—B-type natriuretic peptide; hs—CRP—high-sensitivity C-reactive protein.

**Table 2 jcm-12-02272-t002:** Two-dimensional echocardiographic parameters in NIDCM patients.

2D Echocardiographic Parameters	NIDCM Group (n = 109)
IVS, mm	9.7 ± 1.2
PW, mm	9.6 ± 1.3
GRV, mL	49.5 ± 32.8
Left ventricle	
LVESDi, mm/m^2^	27.6 ± 4.5
LVEDDi, mm/m^2^	32.9 ± 4.1
LVEDVi, ml/m^2^	114.3 ± 37.6
LVESVi, ml/m^2^	82.1 ± 37.0
LVEF, %	27.7 ± 8.7
LVGLS, %	−8.6 ± 2.8
LVGCS, %	−14.1 ± 4.9
LVGRS, %	20.4 ± 9.3
Right ventricle	
RVEDVi, mL/m^2^	71.0 ± 26.6
RVESVi, mL/m^2^	44.5 ± 21.3
RVFWLS, %	−18.2 ± 3.0
RVGLS, %	−16.3 ± 2.4
Left atrium	
LA, mm	45.6 ± 8.5
LAV, mL	115.7 ± 63.6
LAVi, mL/m^2^	56.1 ± 29.3
LAScd, %	−13.4 ± 4.5
LASr, %	23.3 ± 7.2
LASct, %	−9.9 ± 4.3
Right atrium	
RAV, mL	89.8 ± 24.5
RAVi, mL/m^2^	39.9 ± 10.2
RAScd, %	−15.6 ± 5.5
RASr, %	20.9 ± 6.2
RASct, %	−12.4 ± 5.7

NIDCM—non-ischemic dilated cardiomyopathy; IVS—interventricular septum; PW—posterior wall; GRV—global regurgitation volume; LVESDi—left ventricular end-systolic diameter index; LVEDDi—left ventricular end-diastolic diameter index; LVEF—left ventricular ejection fraction; LVEDVi—left ventricular end-diastolic volume index; LVESVi—left ventricular end-systolic volume index; LVGLS—left ventricular global longitudinal strain; LVGCS—left ventricular global circumferential strain; LVGRS—left ventricular global radial strain; RVEDVi—right ventricular end-diastolic volume index; RVESVi—right ventricular end-systolic volume index; RVFWLS—right ventricular free wall longitudinal strain; RVGLS—right ventricular global longitudinal strain; LA—left atrium; LAV—left atrial volume; LAVi—left atrial volume index; LASr—left atrial strain during the reservoir phase; LAScd—left atrial strain during conduit phase; LASct—left atrial strain during contraction phase; RAV—right atrial volume; RAVi—right atrial volume index; RAScd—right atrial strain during the conduit phase; RASct—right atrial strain during the contraction phase; RASr—right atrial strain during the reservoir phase.

**Table 3 jcm-12-02272-t003:** Myocardial mechanics and morphometric correlations with prognostic factors of HF.

Myocardial Mechanics and Morphometrics				Prognostic Factors of Heart Failure					
	Age	Gender	Smoking	Systolic BP	AF	NYHA Class	6MWT	BNP	LVEF
GRV, mL	R = 0.223, *p* = 0.026	-	-	R = 0.210, *p* = 0.036	-	R = 0.431, *p* < 0.001	R = −0.425, *p* < 0.001	*p* = 0.021	R = −0.457, *p* < 0.001
Left ventricle									
LVEDDi, mm/m^2^	-	-	-	-	-	-	-	-	R = −0.312, *p* = 0.002
LVESDi, mm/m^2^	-	-	-	-	-	-	-	-	R = −0.426, *p* < 0.001
LVEDVi, ml/m^2^	-	-	-	R = −0.248, *p* = 0.013	-	R = 0.209, *p* = 0.037	R = −0.367, *p* < 0.001	-	R = −0.386, *p* < 0.001
LVESVi, ml/m^2^	-	-	-	R−0.436, *p* < 0.001	-	-	-	-	R = −0.491, *p* < 0.001
LVEF, %	-	-	R = −0.260, *p* = 0.009	-	-	R = −0.499, *p* < 0.001	R = 0.462, *p* < 0.001	R = −0.461, *p* < 0.001	1
LVGLS, %	-	-	-	-	-	R = 0.384, *p* < 0.001	R = −0.389, *p* < 0.001	R = 0.426, *p* = 0.025	R−0.797, *p* < 0.001
LVGCS, %	-	-	-	-	-	R = 0.343, *p* = 0.014	-	-	R−0.759, *p* < 0.001
LVGRS, %	-	R = 0.285, *p* = 0.004	-	-	-	R = −0.393, *p* < 0.001	R = 0.395, *p* < 0.001	-	R = 0.725, *p* < 0.001
Right ventricle									
RVEDVi, mL/m^2^	-	-	-	-	-	-	-	-	R = −0.332, *p* < 0.001
RVESVi, mL/m^2^	-	R = −0.209, *p* = 0.048	-	-	-	-	-	-	R = −0.340, *p* < 0.001
RVFWLS, %	-	-	-	-	-	R = 0.345, *p* < 0.001	R = −0.356, *p* < 0.001	R = −0.304, *p* = 0.032	R = −0.397, *p* < 0.001
RVGLS, %	-	-	-	-	-	R = 0.345, *p* < 0.001	R = −0.354, *p* < 0.001	-	R = −0.424, *p* < 0.001
Left atrium									
LAVi, mL/m^2^	R = 0.222, *p* = 0.026	-	-	-	R = −0.346, *p* < 0.001	R = 0.210, *p* = 0.036	-	-	R = −0.297, *p* = 0.003
LAScd, %	-	-	-	-	R = −0.392, *p* < 0.001	R = 0.341, *p* = 0.001	-	-	R = −0.835, *p* < 0.001
LASct, %	-	-	-	-	R = −0.402, *p* < 0.001	R = 0.266, *p* = 0.007	-	-	R = −0.567, *p* < 0.001
LASr, %	-	-	-	-	R = −0.434, *p* < 0.001	R = −0.392, *p* = 0.001	-	R = 0.348, *p* = 0.036	R = 0.781, *p* < 0.001
Right atrium									
RAVi, mL/m^2^	-	-	-	-	-	-	-	-	R = −0.281, *p* = 0.005
RAScd, %	-	-	-	-	-	R = 0.314, *p* = 0.001	R = −0.370, *p* < 0.001	-	R−0.494, *p* < 0.001
RASct, %	-	-	-	-	-	R = 0.286, *p* = 0.004	-	-	R = −0.315, *p* = 0.001
RASr, %	-	-	-	-	-	R = −0.322, *p* = 0.001	-	-	R = 0.363, *p* < 0.001

Sign “-“—there are no correlations between parameters; GRV—global regurgitation volume; LVEF—left ventricular ejection fraction; LVESDi—left ventricular end-systolic diameter index; LVEDDi—left ventricular end-diastolic diameter index; LVEDVi—left ventricular end-diastolic volume index; LVESVi—left ventricular end-systolic volume index; LVGCS—left ventricular global circumferential strain; LVGRS—left ventricular global radial strain; LVGLS—left ventricular global longitudinal strain; RVEDVi—right ventricular end-diastolic volume index; RVESVi—right ventricular end-systolic volume index; RVGLS—right ventricular global longitudinal strain; RVFWLS—right ventricular free wall longitudinal strain; LAVi—left atrial volume index; LAScd—left atrial strain during the conduit phase; LASct—left atrial strain during the contraction phase, LASr—left atrial strain during the reservoir phase; RAVi—right atrial volume index; RAScd—right atrial strain during the conduit phase; RASct—right atrial strain during the contraction phase; RASr—right atrial strain during the reservoir phase; BMI—body mass index; BP—blood pressure; AF—atrial fibrillation; NYHA—New York Heart Association; 6MWT—6 min walk test; BNP—B-type natriuretic peptide.

**Table 4 jcm-12-02272-t004:** Parameters correlating with prognostic factors of HF.

Variable		Multivariate Analysis	
	Standardized Coefficient B	95% CI	*p*
Systolic blood pressure			
LVESVi, mL/m^2^	−0.248	−0.226–(−0.028)	0.013
NYHA class			
GRV, mL	0.273	0.002–0.020	0.018
LASr, %	−0.404	−0.011–0.065	0.006
LVEF, %	−0.572	−0.064–(−0.028)	<0.001
RVFWLS, %	0.192	0.002–0.086	0.049
6MWT			
GRV, mL	−0.300	−0.010–(−0.002)	0.005
LVEF, %	0.743	0.033–0.077	<0.001
RVFWLS, %	−0.247	−0.091–(−0.013)	0.009
BNP			
LVEF, %	−0.251	−85.2–(−10.533)	0.013
LVEF			
LVGLS, %	−0.291	−1.282–(−0.526)	<0.001
LVGRS, %	0.207	0.081–0.308	<0.001
GRV, mL	−0.238	−0.758–(−0.167)	0.003
LASr, %	0.267	0.166–0.478	<0.001

CI—confidence interval; 6MWT—6 min walk test; NYHA—New York Heart Association; BNP—B-type natriuretic peptide; LVEF—left ventricular ejection fraction; LVESVi—left ventricular end-systolic volume index; LVGRS—left ventricular global radial strain; GRV—global regurgitation volume; RVFWLS—right ventricular free wall longitudinal strain; LVGLS—left ventricular global longitudinal strain; LASr—left atrial strain during the reservoir phase.

## Data Availability

Not applicable.

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
