# Peer review of "Relationships of Whole-Heart Myocardial Mechanics and Cardiac Morphometrics by Transthoracic Echocardiography with Main Prognostic Factors of Heart Failure in Non-Ischemic Dilated Cardiomyopathy"

_jcm, 2023, doi:10.3390/jcm12062272_

Round 1

Reviewer 1 Report

The authors describe a new approach for depiction of prognostic factors in patients with dilated cardiomyopathy. They use a "whole- heart myocardial mechanics. Overall, this hypothesis is appealing, still, the presentation of the study leaves some questions:

- what is the "myocardial mechanics and cardica morphometrics" exactly? Please provide a short description in the abstract.

- What is the clinical inplication besides prognosis? How can it support the clinician? 

- Also, there should be an introduction and explanation of myocardial mechanics and cardica morphometrics in the introduction and method section. 

- first paragraph of introduction can be more to the point. 

- Methods:

-- please provide the medication

-- what kind of heart failure patients? HFpEF, HFrEF??,,

-- at what stage were the patients seen? first presentation, later during uptitration? please elude on that.

-- is there any difference seen regarding different stages? Some patients are highly symptomatic or even decompensated. This seems comparing apples to oranges. 

- please describe the method more clearly.

- try to cluster the method section e.g. measures of LV, RV etc.

- how do you handle patients in AF? 

- REsults:

- please add medication

- what about devices? 

- Tables are hard to read. Please try to group measurements of particular compartments. 

- discussion:

- the correlation was rather weak, does it help you a lot? please discuss.

- the last sentence is rather speculative. 

Author Response

Dear reviewer,

We are thankful for your important notes for us to improve our manuscript. We have tried to answer all your questions and comments.

- what is the "myocardial mechanics and cardiac morphometrics" exactly? Please provide a short description in the abstract.

A short description in the abstract about “myocardial mechanics and cardiac morphometrics” was added.

- What is the clinical implication besides prognosis? How can it support the clinician?

Two-dimensional speckle tracking echocardiography is a widely used imaging method in clinical practice which allows to evaluate parameters of myocardial mechanics, which correlate with poor prognosis. In this way, it helps to select the highest-risk patients, apply the most appropriate treatment corrections, and thus improve the patient's prognosis. Another important fact is that patients in this group in later period of optimal medical treatment often have implanted devices, which makes echocardiography the first-choice examination method during the follow-up period (the quality of the magnetic resonance examination deteriorates).

- Also, there should be an introduction and explanation of myocardial mechanics and cardiac morphometrics in the introduction and method section.

It was corrected as recommended.

- First paragraph of introduction can be more to the point.

It was corrected as recommended.

- Methods:

-- please provide the medication

Patients did not receive optimal HF treatment prior to enrollment.

-- what kind of heart failure patients? HFpEF, HFrEF??

Patients with HFrEF.

-- at what stage were the patients seen? first presentation, later during uptitration? please elude on that.

The study included ambulatory and hospitalized patients for the first time diagnosed non-ischemic dilated cardiomyopathy (patients without chronic or worsening HF).

-- is there any difference seen regarding different stages? Some patients are highly symptomatic or even decompensated. This seems comparing apples to oranges. 

We did not compare patients with different NYHA class. The study included patients for the first time diagnosed NIDCM.

- please describe the method more clearly.

It was corrected.

- try to cluster the method section e.g. measures of LV, RV etc.

We try to cluster the method section as Your recommended.

- how do you handle patients in AF?

Thank You for Your valuable remarks about method section. We describe the method section more clearly and try to answer Your questions (more details in manuscript).

DCM definition was based according to current criteria as left ventricle (LV) or both ventricles dilatation and systolic dysfunction (LVEF ≤50%) without coronary artery disease (CAD). The study included ambulatory and hospitalized patients for the first time diagnosed NIDCM (patients without chronic or worsening HF). Patients were excluded from the study with significant valve disease, inflammatory myocardial disease, or kidney disease (eGFR <30 ml/min/1.73 m2), tachycardia-induced HF, peripartum cardiomyopathy, with the implantation of intra-cardiac defibrillator or cardiac resynchronization therapy, in case of alcohol or drug abuse and under the age of 18.

We excluded patients with tachycardia-induced cardiomyopathy. The number of patients with atrial fibrillation was counted if there was a history of paroxysmal atrial fibrillation (not chronic normosystolic or prolonged unknown duration tachysystolic form of atrial fibrillation) or atrial fibrillation was detected by Holter monitoring. 

- Results:

- please add medication

Medications were added.

- what about devices?

The patients were excluded from the study if the implantation of an intra-cardiac defibrillator or cardiac resynchronization therapy was in the past.

- Tables are hard to read. Please try to group measurements of particular compartments.

It was corrected.

- discussion:

- the correlation was rather weak, does it help you a lot? please discuss.

The discussion was complemented by a critical approach to these results.

- the last sentence is rather speculative.

It was corrected.

Reviewer 2 Report

Current treatment, medical as well as interventional of the patients should be mentioned:

1) Since there were more patients with NYHA functional classes III-IV (56.9% and 13.8%, respectively), it is likely they were under optimal medical therapy, including possibly diuretics. Initiation and up‐titration of recommended HF therapy in patients might result in significant MR reduction, thus influence GRV and RVFWLS.

2) Did they receive CRT-D since ‘Patients with NIDCM had wide QRS duration (approximately 45% of patients with NIDCM have left bundle branch block)’ – mean LVEF 27.7±8.7 and NYHA III/VI? Again, GRV and RVFWLS will be influenced by CRT.

I do not feel that figure 1 brings any additive value.

Page 4 – there should be: (BMI 29.1…)

In results section (page 4) the pharse ‘AF was present in 39.4% and ventricular tachycardia was present in 30.3% of patients’ is unclear. Were 39.4% in AFib at the moment of inclusion? Did they have history of sustained VT?

In conclusions section, I think the first phrase ’The assessment of whole-heart myocardial mechanics with cardiac morphometric parameters becomes of crucial importance in patients with HF’ is overstated. Focus should be on GRV and RVFWLS, since LVEF is already an established parameter.

Author Response

Dear reviewer,

We are thankful for your important notes for us to improve our manuscript. We have tried to answer all your questions and comments.

1) Since there were more patients with NYHA functional classes III-IV (56.9% and 13.8%, respectively), it is likely they were under optimal medical therapy, including possibly diuretics. Initiation and up‐titration of recommended HF therapy in patients might result in significant MR reduction, thus influence GRV and RVFWLS.

Thank You for Your comment. We tried to supplement our method section and provide more details.

The study included ambulatory and hospitalized patients for the first time diagnosed NIDCM (patients without chronic or worsening HF).

The effect of optimal HF treatment could be evaluated in follow-up.

2) Did they receive CRT-D since ‘Patients with NIDCM had wide QRS duration (approximately 45% of patients with NIDCM have left bundle branch block)’ – mean LVEF 27.7±8.7 and NYHA III/VI? Again, GRV and RVFWLS will be influenced by CRT.

We provided a more detailed method section. Patients with the implantation of an intra-cardiac defibrillator or cardiac resynchronization therapy weren‘t included in the study.

The effect of interventional HF treatment could be evaluated in follow-up.

I do not feel that figure 1 brings any additive value.

We agree with You and it was removed.

Page 4 – there should be: (BMI 29.1…)

It was corrected.

In results section (page 4) the pharse ‘AF was present in 39.4% and ventricular tachycardia was present in 30.3% of patients’ is unclear. Were 39.4% in AFib at the moment of inclusion? Did they have history of sustained VT?

Thank You for Your remark. We refined these results in the text. We excluded patients with tachycardia-induced cardiomyopathy. The number of patients with atrial fibrillation was counted if there was a history of paroxysmal atrial fibrillation (not chronic or prolonged unknown duration tachysystolic form of atrial fibrillation) or atrial fibrillation was detected by Holter monitoring. They didn‘t have a history of sustained VT. VT was detected by Holter monitoring during the first time diagnosed NIDCM.

In conclusions section, I think the first phrase ’The assessment of whole-heart myocardial mechanics with cardiac morphometric parameters becomes of crucial importance in patients with HF’ is overstated. Focus should be on GRV and RVFWLS, since LVEF is already an established parameter.

Thank You. It was corrected as recommended.

Round 2

Reviewer 1 Report

The revised version has improved and the authors have taken my suggestions into accout. Still, I am sorry to say, there remain a few questions:

- a what timepoint during the course of disease were the echos made?

- - Early presentation, treated, during decompensation? this is important to know. 

-- Again, please stratify according to EF in HFpEF, HFrEF and HFmrEF. Do you see any signal if you look at the groups apart? 

- you are writing about myocardial mechanics. I still do not totally understand the aproach. Is it just a generalizing term or a statistical combination/ index of several measurements? 

- I am still struggling with the clinical impact. Does it really help you to add additional measurements that are poorly correlated? Do you expect colleagues to do so? Can you define what the clear benefit is in clinical practice? 

Author Response

Dear reviewer,

Thank you again for Your comments and Your time. We will try to answer your questions.

- a what timepoint during the course of disease were the echos made?

Thank you for Your question. Echocardiography was performed during the first contact with the patient - within 24 hours of the start of hospitalization or if it was an outpatient - during the first visit.

- - Early presentation, treated, during decompensation? this is important to know. 

Yes, it is important fact, we agree. All patients didn’t receive optimal HF treatment before the first echocardiography. We would like to emphasize that the echocardiography was performed during the first contact with the patient. We also mentioned this in the revised methodological part.

-- Again, please stratify according to EF in HFpEF, HFrEF and HFmrEF. Do you see any signal if you look at the groups apart? 

As we mentioned earlier, only patients with reduced LV systolic function were included in the study. Patients with HFpEF were excluded. The average of EF was 27.7±8.7. We didn’t compare the patients with each other according to the LVEF, because most of the patients were with HFmrEF. We try to group patients according to LVEF.

HFrEF (<=40%), n (%)  92 (84.4)

HFmrEF (41-49), n (%) 17 (15.6)

- you are writing about myocardial mechanics. I still do not totally understand the aproach. Is it just a generalizing term or a statistical combination/ index of several measurements? 

Myocardial mechanics are characterized by myocardial deformation parameters (parameters presented in our results - for example, left ventricular global longitudinal strain, left ventricular global circumferential strain; left ventricular global radial strain, right ventricular free wall longitudinal strain, right ventricular global longitudinal strain, etc.).

It is just a generalizing term.

- I am still struggling with the clinical impact. Does it really help you to add additional measurements that are poorly correlated? Do you expect colleagues to do so? Can you define what the clear benefit is in clinical practice? 

Thank You for the question. This discussion is very valuable.

As we mentioned in the discussion section, we obtained weak-moderately strong correlations, but they were statistically significant. The speckle-tracking 2D echocardiography analysis is the gold standard to evaluate myocardial deformation (mechanics). Our results showed that combined with LVEF, GRV, and RVFWLS may add additional clinical value to select the highest-risk patients in clinical practice.  GRV is a new echocardiographic parameter, we haven’t yet used it in practice. Since we obtained statistically reliable results, perhaps the use of these parameters will help select patients who are at the highest risk. We critically evaluated the small sample size; further research is needed to confirm our findings which could help in patient management to improve outcomes.

Thank You again for discussion.

Sincerely,

Authors.

Reviewer 2 Report

Dear authors,

Thank you for your efforts.

I feel that the extensive modifications have made the message clearer and more valuable for the readers.

Author Response

Dear reviewer,

Thank You for Your time and remarks.

The manuscript was checked by a native English-speaking colleague.

Sincerely,

Authors